# Area-under-the-Curve-Based Mycophenolate Mofetil Dosage May Contribute to Decrease the Incidence of Graft-versus-Host Disease after Allogeneic Hematopoietic Cell Transplantation in Pediatric Patients

**DOI:** 10.3390/jcm10030406

**Published:** 2021-01-21

**Authors:** Giorgia Carlone, Roberto Simeone, Massimo Baraldo, Alessandra Maestro, Davide Zanon, Egidio Barbi, Natalia Maximova

**Affiliations:** 1Department of Medicine, Surgery and Health Sciences, University of Trieste, Piazzale Europa 1, 34127 Trieste, Italy; giorgiacarlone@gmail.com (G.C.); egidio.barbi@burlo.trieste.it (E.B.); 2Department of Transfusion Medicine, ASUGI, Piazza dell’Ospitale 1, 34129 Trieste, Italy; roberto.simeone@asugi.sanita.fvg.it; 3Department of Medicine (DAME), University of Udine, Via Colugna 50, 33100 Udine, Italy; massimo.baraldo@uniud.it; 4SOC of Clinical Pharmacology Institute, Friuli Centrale University Hospital, Via Pozzuolo 330, 33100 Udine, Italy; 5Institute for Maternal and Child Health-IRCCS Burlo Garofolo, Via dell’Istria 65/1, 34137 Trieste, Italy; alessandra.maestro@burlo.trieste.it (A.M.); davide.zanon@burlo.trieste.it (D.Z.)

**Keywords:** mycophenolate mofetil, graft-versus-host disease, therapeutic drug monitoring, hematopoietic stem cell transplantation, pediatric patients

## Abstract

Acute graft-versus-host disease (GvHD) remains the second leading cause of death, after disease relapse, in patients undergoing allogeneic hematopoietic stem cell transplantation (allo-HSCT). The medical records of 112 pediatric patients who underwent allo-HSCT from matched unrelated and haploidentical donors were analyzed. Patients were divided into two groups, according to the GvHD prophylactic regimen used. In the control group, GvHD prophylaxis consisted of cyclosporine A (CsA) and methotrexate (MTX) or CsA and mycophenolate mofetil (MMF) at a standard daily dose of 30 mg/kg. All subjects in the study group received tacrolimus (FK506) and MMF. In this group, MMF was subjected to therapeutic drug monitoring (TDM) through mycophenolic acid (MPA) area under the curve AUC_0–12_. We found a statistically significant difference in both overall acute GvHD (*p* < 0.0001) and overall chronic GvHD (*p* < 0.05) incidence between the study and the control group. The initial daily MMF dose and the age at transplant in the study group proved to be inversely correlated (r = −0.523, *p* < 0.0001). The children under six years of age required a significantly higher daily MMF dose (*p* < 0.008). This study showed that pharmacological monitoring of MPA AUC_0–12_ concentration allowed a reduction in the incidence of acute and chronic GvHD. MMF showed age-dependent pharmacokinetics due to greater drug clearance in younger children.

## 1. Introduction

Allogeneic hematopoietic stem cell transplantation (allo-HSCT) has increasingly emerged as a therapeutic option for pediatric patients with hematologic malignancies and non-malignant diseases. Despite pivotal advances in the development of modern transplant procedures, acute graft-versus-host disease (GvHD) remains the second leading cause of death, after disease relapse, in patients undergoing allo-HSCT [1]. Current rates of grade II–IV acute and chronic GvHD after human leukocyte antigen (HLA)-matched related (MRD) and matched unrelated (MUD) transplants are 35–55%, while haploidentical HSCT was initially associated with high incidences of acute and chronic GvHD [2,3]. However, following the implementation of high-dose post-transplantation cyclophosphamide, studies have demonstrated no difference between haploidentical and HLA-matched HSCT with respect to the incidence of acute GvHD, and also confirmed a lower incidence of chronic GvHD [4,5].

Many prophylactic regimens were tested to decrease the incidence of acute GvHD. Among these therapeutic approaches, the combination of cyclosporine A (CsA) or tacrolimus (FK506) with short-course methotrexate (MTX) has become the gold standard prophylaxis [2,5,6]. Furthermore, other immunosuppressant drugs, such as corticosteroids, anti-thymocyte globulin, and anti-CD52 monoclonal antibody alemtuzumab, have been used either as alternative drugs or as a supplemental therapy to standard prophylactic treatment [5].

In the last two decades, mycophenolate mofetil (MMF), an ester prodrug of mycophenolic acid (MPA), which acts by blocking the de novo purine synthesis of T- and B-cell lymphocytes, has been developed with different prophylactic regimens to obtain effective immunosuppression [5,7]. With respect to MTX, the MMF seems to cause less cytotoxicity, with a smaller incidence of severe mucositis and related supportive care, and it also enables faster engraftment [5,8]. Available data on the pharmacokinetics of MPA show significant inter- and intra-patient variation. However, several studies have demonstrated that the relatively higher area under the curve (AUC) of the MPA determines significant suppression of acute GvHD in a prophylactic regimen. Still, there are few data available on the pediatric population [5].

Our study aimed to determine whether FK506 plus MMF prophylactic regimen, with MMF administered dose based on AUC measurement of MPA, contributed to decrease the incidence of acute and chronic GvHD in pediatric patients undergoing allo-HSCT.

## 2. Materials and Methods

### 2.1. Study Design and Population

A retrospective single-center study was carried out at the Pediatric Transplant Center of the Institute for Maternal and Child Health-IRCCS Burlo Garofolo, Trieste, Italy. The Institutional Review Board and Ethics Committee of the IRCCS Burlo Garofolo approved the study protocol (reference no. 2275/2015). All parents of the patients gave written consent to collect and use personal data for research purposes. The medical records of all patients who underwent allo-HSCT from matched unrelated and haploidentical donors at our center between January 2002 and June 2019 were analyzed. The current study’s inclusion criteria were: age of recipient at transplantation under 18 years, first transplant attempt, myeloablative conditioning regimen, and a minimum 12-month follow-up period after HSCT.

Data were analyzed for various demographic, clinical, and HSCT variables, including gender, age, underlying disease, conditioning regimen, donor type, graft source, graft cellular composition, and GvHD prophylaxis. The disease risk was established, considering the primary diagnosis and the disease stage [9]. Regarding the GvHD prophylactic regimen, two different cohorts were described in this study: the historical or ‘control group’, which included patients undergoing HSCT between January 2002 and December 2010, and the study group of pediatric patients experiencing allo-HSCT between January 2011 and December 2019. In addition, the following data were collected for patients enrolled in the study group: initial and final MMF daily dose, duration of MMF prophylaxis, the value of first and last AUC measurement of MPA, number of AUCs performed, the achievement of target MPA values, and timing of leukocyte and neutrophil engraftment. The primary outcome evaluated was the possible difference in the incidence of acute and chronic GvHD in both groups during a 12-month follow-up period. Assessed secondarily was the relationship between pharmacokinetics of MMF and the age of transplant recipients, particularly the daily dose of MMF needed to achieve the MPA target concentration and the correlation between MPA AUC values and time of myeloid engraftment.

### 2.2. HSCT Procedure

Patients were treated according to the standard therapy and local protocols derived from international guidelines. The conditioning regimen was myeloablative in all subjects and differed based on the primary disease. In most cases, it consisted of busulfan or fractioned total body irradiation with the addition of thiotepa and cyclophosphamide, as previously described [10]. In the event of MUD and haploidentical donors or sickle cell anemia as a primary diagnosis, rabbit anti-thymocyte globulin (ATG) was used, specifically Thymoglobulin^®^, Genzyme.

### 2.3. GvHD Prophylaxis

Patients enrolled in the study were divided into two groups, according to the GvHD prophylactic regimen used. In the control group, GvHD prophylaxis consisted of CsA and a short-course of MTX or CsA and oral MMF, type CellCept^®^ (Roche Pharma AG, Emil-Barell-Str. 1, 79639 Grenzach-Wyhlen, Germania)**,** at a standard daily dose of 30 mg/kg with pre-dose therapeutic drug monitoring (TDM). All subjects in the study group received FK506 and MMF as GvHD prophylaxis. Specifically, CsA was administered intravenously at a dose of 1 to 3 mg/kg/d for the first 21 days and subsequently orally at a dose of 6 mg/kg/d. Initial target trough level was set at 250–350 ng/mL and, later, during oral administration, at 200–250 ng/mL. On the other hand, FK506 was administered intravenously at a dose of 0.03 mg/kg/d with initial target concentration setting at 15–20 ng/mL, while, during oral administration, the target trough level was set at 10–15 ng/mL.

Patient variables, such as primary disease, disease risk at transplant, infection risk; and transplant-related variables, such as donor type, graft source, graft composition, are taken into account to calculate the starting MMF dose. Moreover, the starting daily dose is approximately corrected for the patient’s age, according to the ward’s internal protocol (≥50 mg/kg in patients under six years of age, 30–50 mg/kg in older patients). The initial daily dose of MMF in the study group was a minimum of 30 mg/kg and subjected to therapeutic drug monitoring (TDM) through MPA AUC_0–12_.

### 2.4. MPA AUC_0–12_ Monitoring and Dose Tailoring

The blood samples were collected at time 0, 30, and 120 min, respectively. MPA concentration assessment was performed by liquid chromatography-mass spectrometry. The limit of detection (LOD) was 0.4 µg/mL. The method revealed linearity between 0.4 µg/mL and 20 µg/mL (mean correlation coefficient, *R*^²^ = 0.998). Our laboratory found that the intra-day precision (RDS%) and accuracy (%) of the method at MPA plasma concentrations of 0.4, 4.0, 20.0 µg/mL was 6.91% and −0.50%, 6.49% and 0.40% and 5.57% and 0.75%, respectively; the corresponding inter-day precision (RDS%) and accuracy (%) was estimated to be 6.63% and −0.10%, 6.69% and −0.69% and 5.94% and 1.43%, respectively.

The estimated AUC_0–12_ was calculated employing a pharmacokinetics sampling model validated for the kidney transplant (estimated AUC_0–12_ = 18.6 + 4.3 × C0 + 0.54 × C0.5 + 2.15 × C2) [11]. AUC_0–12_ target range was considered from 30 to 60 μg/mL/h, adopting the target followed in a previous study [11]. Starting from the MMF initial daily dose, and tailoring MMF dosage based on the first AUC_0–12_, thereafter we performed AUC measurement of MPA weekly and we changed MMF dose until AUC measurement of MPA was within the defined range 30–60 µg/mL/h.

### 2.5. Statistical Analysis

Categorical variables were expressed as absolute value and percentage, whereas quantitative variables were reported using the median and interquartile range. Patients’ demographic and clinical characteristics were compared using the chi-square test or Fisher’s exact test for categorical variables, whereas the Mann–Whitney rank-sum test was used for continuous variables. The primary endpoint, acute and chronic GvHD incidence in the study and the control group after a 12-month follow-up period, was calculated according to the Kaplan–Meier method. *p* < 0.05 was considered to be statistically significant.

Data were analyzed using WinStat (v.2012.1; In der Breite 30, 79189 Bad Krozingen, Germany) and MedCalc (Statistical Software version 18.9.1, Ostend, Belgium; http://www.medcalc.org; 2018).

## 3. Results

### 3.1. Study Population and Transplant Characteristics

Our cohort consisted of 76 patients: the study group included 51 subjects, whereas the historical group consisted of 25 patients. The demographics, clinical, and HSCT features of the 76 patients are summarized in Table 1.

The underlying diseases were malignant in 45 children (82%) versus 46 (81%) in the study group and the historical group, respectively. In the study group, the donor’s type was a matched unrelated in 47 cases (85%), and a haploidentical in the remaining eight (15%). In the historical group, a matched unrelated donor was used in 45 cases (79%), and a haploidentical donor in 12 patients (21%). The graft source was bone marrow in 34 children (62%) versus 44 (77%) and peripheral blood stem cells (PBSC) in the remaining 21 subjects (38%) versus 13 patients (23%) in the study group and the historical group, respectively. The median CD34+/kg of recipient weight dose for PBSC was 6.65 × 10^6^ in the study group versus 5.79 × 10^6^ in the historical group. The median total nuclear cells/kg for bone marrow was 5.42 × 10^8^ and 4.34 × 10^8^, respectively.

### 3.2. Primary Outcome: 12-Month Acute and Chronic GvHD Incidence

After a 12-month follow-up period, we observed a statistically significant difference in both overall acute GvHD (*p* < 0.0001) and overall chronic GvHD (*p* < 0.05) incidence between the study and the historical group. Eighteen patients (72%) from the historical group had at least one episode of acute GvHD (any grade) versus seven patients (13.7%) in the study group.

Comparing the incidence of different grades of acute GvHD in the study and the historical group, we found statistically significant differences in the rate I-II grade GvHD and III-IV grade GvHD (*p* < 0.0001 and *p* < 0.001, respectively) (Figure 1).

### 3.3. Secondary Outcomes

We did not find statistically significant differences in overall survival (OS) and progression-free survival (PFS) between the two groups (*p* > 0.05). Estimated overall survival (OS) in the study group at 1 and 2 years post-allo-HSCT was 80% and 76.4%, respectively. In the historical group, OS was 76% and 72% at 1 and 2 years post-allo-HSCT (Figure 2). Progression-free survivals (PFS) in the study group at 1 and 2 years post-allo-HSCT was 74.5% and 68.6%, respectively. PFS in the historical group was 64% and 56%, respectively, at 1 and 2 years post-transplantation (*p* > 0.05).

The non-GvHD-related mortality was 10.9% and 8.8% in the study and the historical group, respectively (*p* > 0.05). Finally, we compared the incidence of malignancy relapses in both groups, finding it similar. Specifically, eight patients (15.7%) relapsed in the study group, versus five patients (20%) in the historical group (Figure 3).

Comparing the two groups, the incidence and type of infections were similar, except for the BK virus (BKV) infection. In the study group, the incidence of BKV reactivation was significantly higher (72.5% versus 32% in the study and the historical group, respectively (*p* < 0.05)).

We compared the distribution of the last MPA AUC_0–12_ values in patients with acute GvHD and patients without acute GvHD. The mean of MPA AUC_0–12_ in patients with GvHD was 30.2 μg h/mL (SD ± 6.8 μg h/mL) versus 39.7 μg h/mL (SD ± 27.3 μg h/mL) without GvHD. The graphical distribution of the last MPA AUC_0–12_ values in both groups through their quartiles are displayed in Figure 4.

We analyzed the relationship between the pharmacokinetics of MMF (MPA AUC_0–12_) and the age of transplant recipients in the study group, proving a strong inverse correlation (*r* = −0.523, *p* < 0.0001) between the initial daily MMF dose and the age at transplant (Figure 5).

The receiver operating characteristic (ROC) curve analysis showed that the MMF initial daily dose > 51.9 mg/kg was able to predict the age at transplant with 75% sensitivity and 65% specificity [AUC_0–12_: 0.708, 95% CI (0.570–0.823), *p* = 0.0034] (Figure 6A). Similarly, we evaluated the predictive performance of the first MPA AUC_0–12_ (Figure 6B).

We observed that age over six years performed best in predicting the MPA target’s achievement with 72% sensitivity and 62% specificity [AUC_0–12_: 0.651, 95% CI (0.511–0.755), *p* = 0.0462].

The box plot analysis of the initial MMF daily dose and the age at transplant revealed that the children under six years of age required a significantly higher dose (*p* < 0.008). Moreover, comparing the time of myeloid engraftment among the groups of subjects under and over age six, we found a delay in neutrophils but not in total leukocyte engraftment in the group of younger patients (*p* < 0.005). These box-plot analyses are displayed in Figure 7A,B.

Age-related variability of MMF exposure is reported in Table 2.

## 4. Discussion

This study shows that pharmacological monitoring of MPA AUC_0_**_–_**_12_ concentration allows a reduction in the incidence of acute and chronic GvHD after allo-HSCT in patients undergoing prophylactic treatment with FK506 plus MMF. The literature reports many trials performed to identify the effective MMF concentration range for GvHD prophylaxis, which is generally uncertain due to the wide variability of MPA serum levels among different patients as well as within a patient over time [5]. The majority of these clinical trials were conducted to establish MMF pharmacokinetics in patients undergoing a solid organ transplant, especially of the kidney, while only a few studies have involved pediatric patients after allo-HSCT [12].

Our study, suggesting pharmacological monitoring of MMF by calculating MPA AUC_0_**_–_**_12_ to obtain an optimal MMF dosage, demonstrated that patients undergoing prophylactic treatment with FK506 plus MMF showed a significant reduction in the incidence of acute and chronic GvHD after allo-HSCT.

These findings support previous data that have already proved that monitoring MPA AUC_0_**_–_**_12_ or steady-state concentration (Css) seemed to be helpful in evaluating MMF efficacy in GvHD prophylaxis both in allo-HSCT and solid organ transplant recipients [13,14,15]. Among these, in a retrospective study involving 20 pediatric patients undergoing allo-HSCT and prophylactic treatment with MMF, Licata et al. required on average as many as three sampling of AUC_0_**_–_**_12_ to monitor MMF therapeutic range, using the same pharmacokinetics sampling model validated for kidney transplant [16]. At the first assessment, only seven patients (35%) achieved MPA in the therapeutic range for AUC_0_**_–_**_12_, while, at the second monitoring, 17 patients (85%) reached the MPA range [16]. Similarly, in another study, Windreich et al. tested the effect of MMF’s continuous infusion for prophylaxis of GvHD in a cohort of pediatric patients, adjusting the target MMF prophylactic dose through MPA AUC_0–12_ measurement. In conclusion, the authors observed no toxic deaths, excellent engraftment, and low rates of grades III to IV acute and chronic GvHD in the study group [17]. Regarding the literature data, our results appear to be promising, presenting a relevant lower incidence of acute GvHD, of any grade, and chronic GvHD, even if the sizes of the analyzed samples are different (Table 3).

Fixed-dose MMF concentrations vary widely from individual to individual due to differences in MPA’s bioavailability and clearance. Primarily, mucosal damage caused by myeloablative conditioning, total body irradiation (TBI), and broad-spectrum antibiotic treatment is responsible for the drug’s lower oral bioavailability; however, following intravenous administration of MMF, HSCT recipients present MPA plasma concentrations 10 times lower than in healthy volunteers [3,18,19,20,21,22]. Therefore, in addition to MMF bioavailability, MPA clearance is a crucial determinant of the MPA plasma level. With respect to this issue, our analysis demonstrated a strong inverse correlation between initial MMF daily dose and age at transplant (*r* = −0.523, *p* < 0.0001).

Our data revealed that patients under six years of age required a significantly higher initial MMF daily dose than did older children. This difference seemed to be due to greater drug clearance in younger children, highlighting the already known age-dependent pharmacokinetics of MPA. Worth noting, Bhatia et al. showed that children <12 years old have significantly higher weight-adjusted clearance and volume of distribution than those included in the 12–16 years age group after intravenous MMF administration in combination with FK506 [7,13]. Moreover, based on a population pharmacokinetic analysis in HSCT patients, Kim et al. also reported that predicted unbound MPA clearance adjusted by weight was higher in smaller children and declined with increasing body weight [7,23].

Furthermore, we observed delayed neutrophils engraftment, averaging 22 days for patients under six years of age, compared with 16 days for patients over six years of age. This delay seemed to be due to the higher total dose of administered MMF. Our findings are consistent with previous data showing T cell IL-17 suppression after MPA treatment with reduced granulocyte-colony-stimulating factor from bone marrow and the related decrease in peripheral blood neutrophil count [8].

Finally, despite the reduction in the incidence of GvHD, we have not documented any increase in the relapse rate. The percentage of primary disease relapse remained unchanged in the study group, despite a significant GvHD reduction. This observation was unexpected because the protective function of graft-versus-leukemia against relapse is well documented [24,25].

To our knowledge, this is the first study reporting the relevance of MPA AUC_0–12_ monitoring in pediatric patients undergoing allo-HSCT, in order to obtain early the most effective prophylactic dose of MMF, which is clearly influenced by differences in MPA pharmacokinetics between various pediatric age groups. Indeed, the early achievement of tailored MMF dose through MPA AUC_0–12_ monitoring allowed us to obtain better outcomes concerning GvHD incidence. Furthermore, our findings add data to available literature by showing relevant difference in MPA pharmacokinetics between patients under and over six years of age. This new cut-off age should help guide the choice of different starting MMF daily doses, compared with current standards, for GvHD prophylaxis in pediatric patients, in addition to AUC measurement of MPA.

Our results did not demonstrate statistically significant differences in the OS and PFS between the study and the historical group. The cumulative incidence for relapse-related mortality proved similar between the two groups. However, life expectancy in pediatric patients undergoing allo-HSCT is notably greater than adults undergoing the same treatment. Therefore, lowering transplant-related complications, such as GvHD, becomes relevant to improving pediatric patients’ quality of life, by avoiding chronic morbidity.

However, some limitations should be considered. First, this is a retrospective, monocentric study, with a relatively small selected sample. Another bias could be AUC_0–12_, which has been estimated and not calculated. A further issue is the lack of a randomly assigned control group, a choice influenced by ethical issues but nevertheless adding a temporal bias. To help compensate, we have selected a historical group from our own clinic as our control group. This choice introduces the historical differences in anti-infective therapies, selection of donors, and prophylactic regimens for GvHD that could be responsible for the higher detected incidence of acute and chronic GvHD in the historical group versus the study group. The heterogeneity of selected groups allowed us to obtain a larger sample size for statistical analysis, despite known limitations of this choice. Multivariate testing is not feasible for potential confounding effects due to the relatively small size of the sub-cohorts. Further investigations, especially randomized controlled trials, could be useful in fulfilling this study’s goals.

## 5. Conclusions

Our results suggest that monitoring MPA AUC_0–12_ seems to contribute to the decrease in the incidence of acute and chronic GvHD in pediatric patients undergoing allo-HSCT. MPA AUC_0–12_ measurement should be taken into consideration in these patients, as well as tailored initial MMF daily doses with respect to different pediatric age groups, in order to obtain better outcomes through early achievement of the most effective prophylactic dose. Further investigations, especially randomized controlled trials, could be valuable in refining our hypotheses.

## Figures and Tables

**Figure 1 jcm-10-00406-f001:**
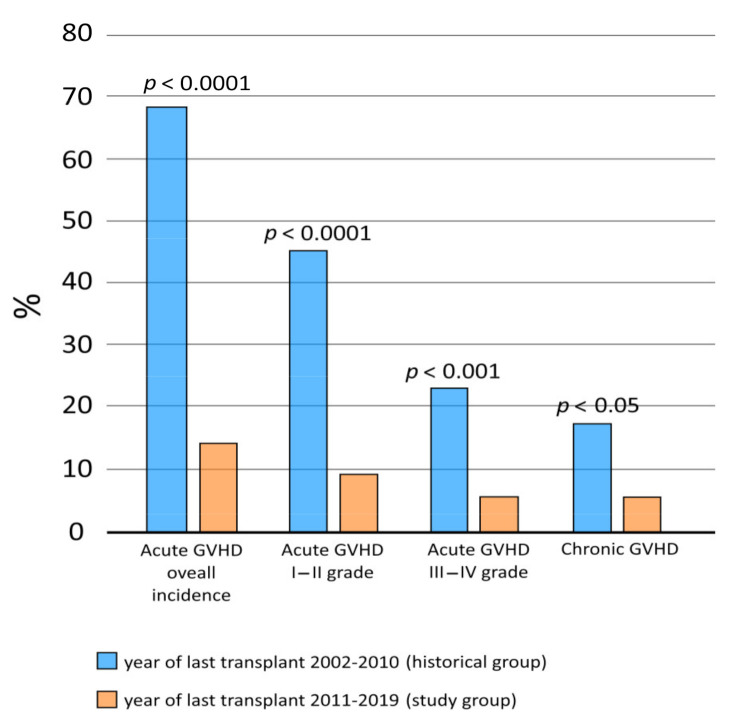
Incidence in the study group and the historical group of overall acute graft-versus-host disease (GvHD), acute GvHD with respect to different grades, and overall chronic GvHD.

**Figure 2 jcm-10-00406-f002:**
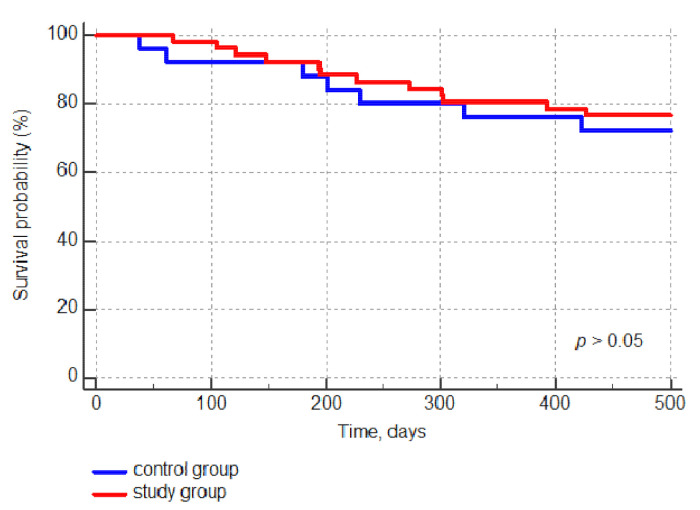
Comparison of the Kaplan–Meier curve for overall survival (OS) in the study group and the control group.

**Figure 3 jcm-10-00406-f003:**
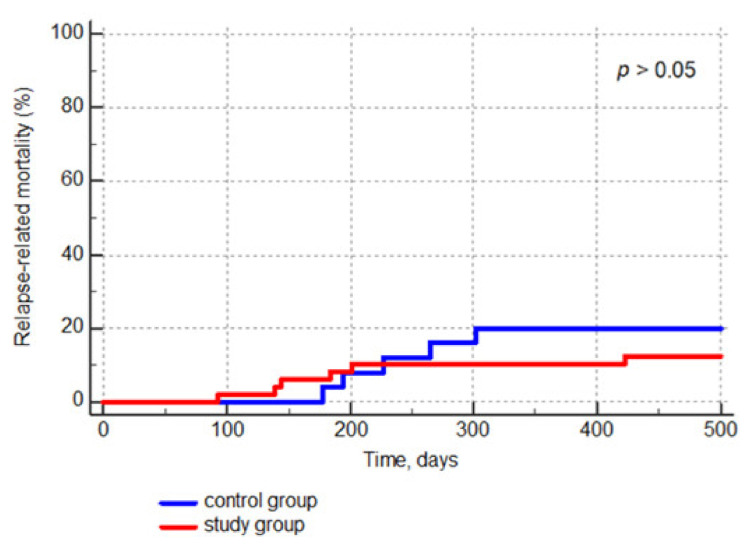
Comparison of the cumulative incidence curve for relapse-related mortality in the study group and the control group.

**Figure 4 jcm-10-00406-f004:**
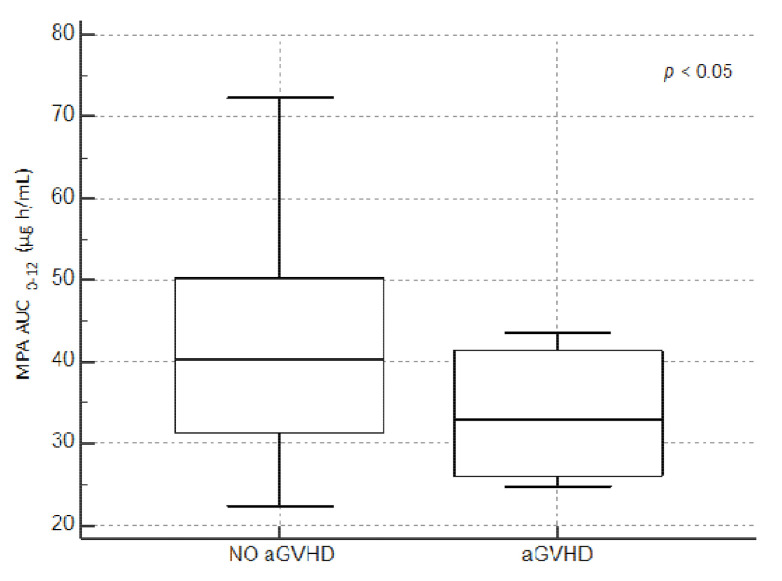
Effect of the last mycophenolic acid (MPA) area under the curve (AUC_0–12_) concentration on the incidence of acute graft-versus-host disease (aGvHD).

**Figure 5 jcm-10-00406-f005:**
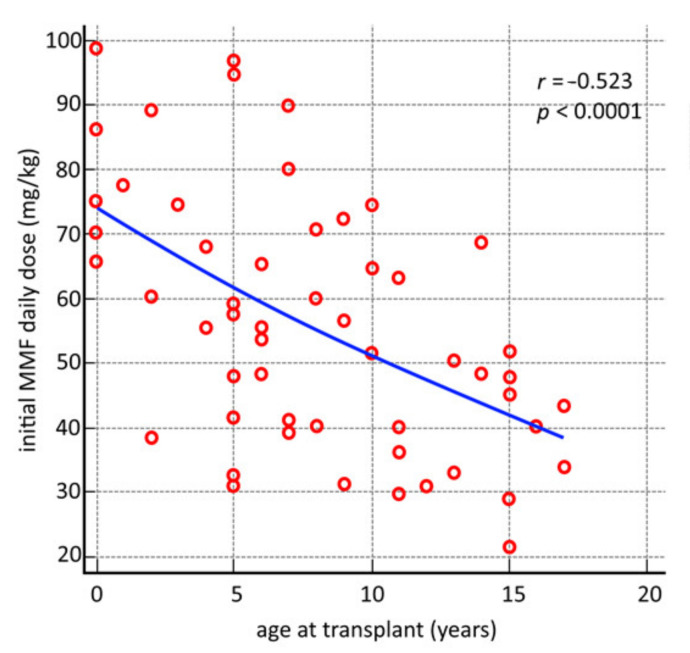
Correlation between initial mycophenolate mofetil (MMF) daily dose and age of transplant recipients.

**Figure 6 jcm-10-00406-f006:**
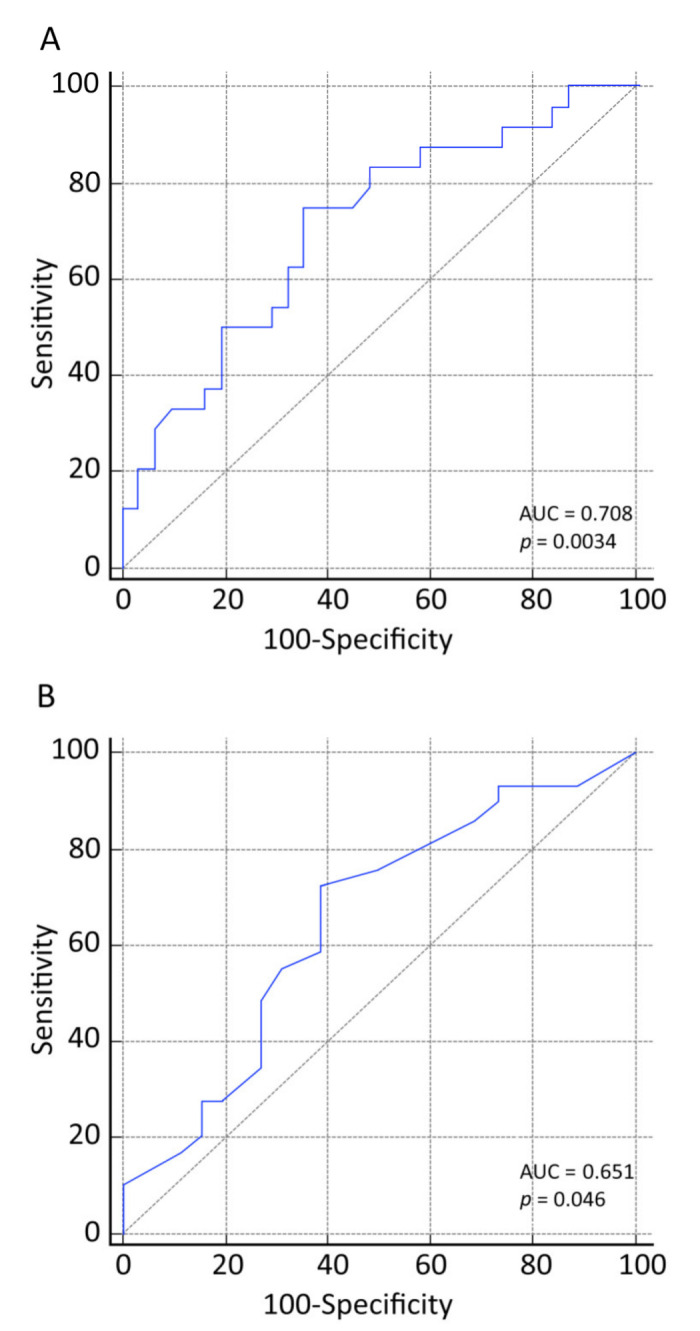
Diagnostic performance of mycophenolate mofetil (MMF) initial daily dose (**A**) and first mycophenolic acid (MPA) area under the curve (AUC)_0–12_ (**B**) in predicting the age at transplant.

**Figure 7 jcm-10-00406-f007:**
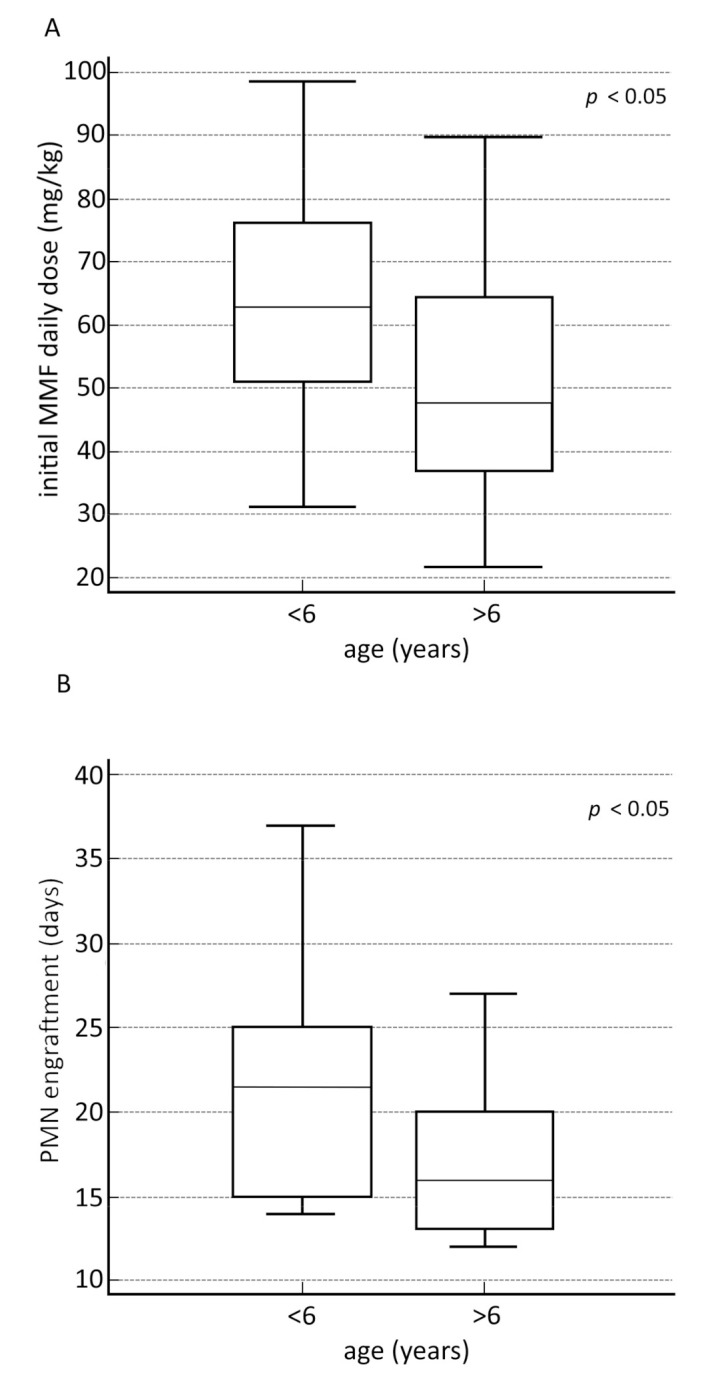
Box-plot analysis of secondary outcomes. Box-plot analysis showed a statistically significant difference between the initial MMF daily dose and the age at transplant, for children above and below the age of six, accounting for a reduced MMF initial daily dose for patients over six years of age (*p* < 0.008) (**A**). A statistically significant difference was found in the time of myeloid engraftment between transplant recipients under or over six years of age (*p* < 0.0049) (**B**).

**Table 1 jcm-10-00406-t001:** Patient demographics, clinical, and HSCT features

Pre-Transplant Baseline Characteristics.	Study Group	Historical Group	*p*-Value
Number of patients (%)	51 (100)	25 (100)	-
Sex, number (%):			
Male	29 (56.9)	16 (57.9)	NS
Female	22 (43.1)	9 (41.1)	
Age at transplant, years, median (IQR)	7 (5–11)	7 (4–12)	NS
Underlying disease, number (%):			
Acute lymphoblastic leukemia	23 (45.1)	11 (44.0)	NS
Acute myeloid leukemia	5 (9.8)	3(12.0)	NS
Myelodysplastic syndrome	11 (21.6)	4 (16.0)	NS
Nonmalignant disorders	9 (17.6)	5 (20.0)	NS
Solid tumor	3 (5.5)	2(8.0)	NS
Disease risk, number (%):			
Standard	27 (49.1)	14 (56.0)	NS
High	24 (47.1)	11 (44.0)	NS
Myeloablative conditioning, number (%):			
MCHT-based	34 (66.7)	16 (64.0)	NS
TBI-based	19 (34.5)	9 (36.0)	NS
Donor type, number (%):			
Matched unrelated donor	47 (92.2)	20 (80.0)	NS
Haploidentical donor	4 (7.8)	5 (20.0)	NS
Female donor to male recipient, number (%)	10 (18.2)	7 (28.0)	NS
Graft source, number (%):			
Bone marrow	35 (68.6)	20 (80.0)	NS
Peripheral blood stem cells	18 (35.3)	5 (20.0)	NS
Graft cellular composition, median (IQR):			
CD 34+ cells × 10^6^/kg	6.32 (4.7–9.31)	5.27 (3.87–7.04)	NS
TNC × 10^8^/kg	5.22 (4.43–8.01)	4.51 (3.95–5.46)	NS
GvHD prophylaxis, number (%):			
Tacrolimus + MMF	47 (92.2)	-	-
Cyclosporin + MTX + MMF	-	23 (92.0)	-
Sirolimus + MMF	4 (7.8)	2 (8.0)	NS

IQR, interquartile range; MCHT, myeloablative chemotherapy; TBI, total body irradiation; CD, cluster of differentiation; TNC, total nuclear cells; MMF, mycophenolate mofetil; MTX, methotrexate; GvHD, graft-versus-host disease.

**Table 2 jcm-10-00406-t002:** Age-related variability of mycophenolate mofetil exposure.

VARIABLES	WHOLE COHORT	AGE 0–6 YEARS	AGE 7–17 YEARS	*p*-Value
Patients, number (%)	55	24 (43.6)	31 (56.4)	-
MMF initial daily dose, mg/kg, median (IQR)	55.5 (40–70.4)	62.9 (49.7–76.8)	47.6 (36.4–64.7)	0.00852
First MPA AUC_0–12_, μg h/mL, median (IQR):	30.4 (26.5–41.2)	28 (24.1–35.6)	32.8 (28.7–43.9)	0.00988
<30, number (%)	26 (47.3)	16 (61.5)	10 (38.5)	0.0152
30–60, number (%)	24 (43.6)	7 (29.2)	17 (70.8)	0.0991
>60, number (%)	5 (9.1)	1 (2)	4 (80)	0.3728
MMF final daily dose, mg/kg, median (IQR)	59.2 (48.1–85.4)	82.1 (56.4–94.1)	52.1 (40.4–68)	0.00048
Last MPA AUC_0–12_, μg h/mL, median (IQR):	40.2 (30.3–47.7)	40 (29.1–44)	40.3 (30.3–49.5)	0.64135
<30, number (%)	13 (23.6)	6 (46.2)	7 (53.8)	1
30–60, number (%)	36 (65.5)	14 (38.9)	22 (61.1)	0.3973
>60, number (%)	6 (10.9)	4 (66.6)	2 (33.4)	0.3866
MMF prophylaxis duration, days, median (IQR)	26 (20–31)	23.5 (19–29.7)	28 (23–31)	0.06637
Number of AUC_0–12_ performed, median (IQR)	2 (2–3)	2 (2–2.7)	2 (2–3)	0.41593
MMF dosage change, number, median (IQR)	1 (1–1)	1 (0–1)	1 (1–1)	0.31518

MMF, mycophenolate mofetil; IQR, interquartile range; MPA, mycophenolic acid; AUC, area under the curve.

**Table 3 jcm-10-00406-t003:** Main characteristics, demographics, and efficacy data of selected studies

Author	Type of Study	Number of Patients	Median Age at Transplant (Years)	GvHD Prophylaxis Regimen with MMF (Dose and Route of Administration)	Acute GvHD Incidence	Moderate or Severe Chronic GvHD	Median Time to Neutrophil Engraftment (Days)
Bolwell, 2004	Randomized controlled	40	49 (19–60)	500 mg 3 times daily either IV or orally	grade I–II: 52% grade II–IV: 48%	63%	11 (range 8–24)
Kiehl, 2002	Randomized controlled, multicenter	45	not available	(1) 1 g twice daily (*n* = 14) (2) 1.5 g twice daily (*n* = 18) IV	(1) grade II–IV: 43% (2) grade II–IV: 66%	not available	not available
Perkins, 2010	Randomized controlled, single center	92	49.9 (23–66.2)	30 mg/kg/ day IV in two divided doses	grade II–IV: 78% grade III–IV: 19%	38%	15
Hamilton, 2014	Retrospective study	241	47 (19–68)	First given at a dose of 500 IV three times a day >increased to 1000 mg twice daily	grade II–IV: 37% grade II–IV: 17%	28%	11
Windreich, 2016	Prospective	19	6 months to 21 years	15 mg/kg every 8 h from day 0 and administered as a constant 2-h infusion > a target total MPA AUC_0–24_ of 40 to 80 µg/mL/h. While on IV CI of MMF, total MPA levels were measured 3 times weekly and MMF dose adjusted to maintain a total MPA Css of 1.7 to 3.3 mg/mL	Six of 18 assessable patients (33%) developed grades II to IV acute GVHD (mean MPA levels were 0.5 mg/mL in those with acute GVHD versus 0.6 mg/ mL in those without acute GVHD (*p* 0.40).	Two of 15 developed moderate severity chronic GVHD	18 of 19 patients (95%): median of 13 days (range, 8 to 41) - BMT: engrafted at 11 days (range, 8 to 43); - CB: 17 days (range, 15 to 41) (*p* < 0.01)

IV, intravenous; CI, continuous infusion; Css, concentration at steady state; MMF, mycophenolate mofetil; MPA, mycophenolic acid; GvHD, graft-versus-host disease; BMT, bone marrow transplantation; CB, cord blood. Variability in serum concentrations of MPA is another issue to consider. MPA serum levels are lower in patients undergoing allo-HSCT compared with organ transplant recipients, specifically subjects undergoing kidney transplants, where the recommended therapeutic range for MPA AUC_0–12_ remains 30–60 µg/mL/h [18].

## Data Availability

The data presented in this study are available on request from the corresponding author.

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
