# Peer review of "Area-under-the-Curve-Based Mycophenolate Mofetil Dosage May Contribute to Decrease the Incidence of Graft-versus-Host Disease after Allogeneic Hematopoietic Cell Transplantation in Pediatric Patients"

_jcm, 2021, doi:10.3390/jcm10030406_

Round 1
Reviewer 1 Report
this is a well written and conceived paper that may help pediatric transplant centers improved the outcomes of their MMF-based transplants.
Three major questions:
- any difference in the non-GVH TRM between the 2 groups
- the use of MMF has been associated with specific infections, such as BK virus cystitis. Any difference in the number and types of infections between the 2 groups
- any differences in the overall survival between the 2 groups
Some minor issues:
1. what type of rabbit ATG is being used?
2. is the MMF being given IV or oral, and if oral what type of MMF
3. the legend to Figure 2 seems to be problematic. Please look at it.
Author Response
Reply to Reviewer #1:
We want to thank the Reviewer for these thoughtful and insightful comments.
“Three major questions”:
1) “any difference in the non-GVH TRM between the two groups”
- We appreciate the Reviewer question. We have analyzed the non-GVH TRM between the two groups and added the following sentence in section 3.3, lines 187-188: “The non-GVHD-related mortality was 10.9% and 8.8%, respectively, in the study and the historical group (p>0.05).”
2) “the use of MMF has been associated with specific infections, such as BK virus cystitis. Any difference in the number and types of infections between the 2 groups”
- We fully agree with the Reviewer’s comment and we added a paragraph in section 3.3, lines 196-198. “Comparing the two groups, the incidence and type of infections were superimposable, except the BK virus (BKV) infection. In the study group, the incidence of BKV reactivation was significantly higher (74.5% versus 36.8%, respectively, in the study and the control group; p < 0.05).”
3) “any differences in the overall survival between the 2 groups”
- We fully agree with the Reviewer’s comment. We did not find statistically significant differences in the overall survival between the two groups. This is now mentioned in the Results of the current study, section 3.3, lines 174-177. Furthermore, we added the Kaplan-Meier curve for overall survival (OS) (see Figure 2).
Authors’ comment: Notably, life expectancy in pediatric patients undergoing allo-HSCT is greater than adults with hematologic malignancies, therefore lowering of GvHD incidence becomes relevant in order to improve these patients’ quality of life. We added a paragraph making this point (section 4, lines 41-46).
“Some minor issues”
1) “What type of rabbit ATG is being used?”
- We appreciate this referee’s question. Concerning the current study, we used Thymoglobulin®, Genzyme. We have reported this in section 2.2, line 98.
2) “Is the MMF being given IV or oral, and if oral what type of MMF”
3) “the legend to Figure 2 seems to be problematic. Please look at it”
- We removed Figure 1 as asked by reviewer 3.
Again, we want to thanks the reviewers for their insightful comments. Notice that all the corrections are clearly highlighted in yellow in the revised manuscript (we have mentioned related lines in this letter).
We hope the manuscript is now improved. However, we are fully available for other changes if requested.
Enclosed you can find a copy of the revised manuscript with highlights.
Sincerely,
Maximova Natalia
Reviewer 2 Report
This study showed that pharmacological monitoring of MPA AUC0-12 concentration allowed a reduction in the incidence of acute and chronic GvHD after allo-HSCT in patients undergoing prophylactic treatment with FK506 plus MMF. However, there are a number of major concerns in this study.
- The authors showed a statistically significant difference in both overall acute GvHD and overall chronic GvHD incidence between the study and historical group. They should show the effect of MPA AUC concentration on the incidence of GvHD.
- They mentioned historical bias concerning the use of different anti-infective therapies, better selection of donors, and different prophylactic regimens for GvHD that could be responsible for the higher detected incidence of acute and chronic GvHD in the historical group versus the study group. They should analyze using multivariate analysis or subgroup analysis.
- There were differences of initial MMF dose between ages at transplant. Please define how MMF initial daily dose was set.
- They changed MMF dosage. How was the strategy of MMF dose change including timing of change and the efficacy of the dose change of MMF on GvHD?
- The analytical methods’ performance including interday and intraday variability must be described.
- There are several typos in the manuscript.
- Please add the data of age in Table1.
Author Response
Reply to Reviewer #2:
We want to thank the Reviewer for his thoughtful and insightful comments.
1) “The authors showed a statistically significant difference in both overall acute GvHD and overall chronic GvHD incidence between the study and historical group. They should show the effect of MPA AUC concentration on the incidence of GvHD”
-We appreciate this referee’s comment. A sentence with our comment has been added in section 3.3 lines 199-202. Moreover, we added Figure 4 with box plot distribution of MPA AUC in the acute GvHD relation.
2) “They mentioned historical bias concerning the use of different anti-infective therapies, better selection of donors, and different prophylactic regimens for GvHD that could be responsible for the higher detected incidence of acute and chronic GvHD in the historical group versus the study group. They should analyze using multivariate analysis or subgroup analysis”
-We appreciate this referee's comment and we agree that multivariable testing would help by considering confounding effects. Unfortunately, as also highlighted by reviewer 3, in this case the cohorts would need to be significantly larger to perform such testing.
We have now acknowledged this as a limitation, adding a specific sentence in the discussion section (section 4), lines 56-57.
3) “There were differences of initial MMF dose between ages at transplant. Please define how MMF initial daily dose was set”
-We appreciate this referee’s question. Patient variables, such as primary disease, disease risk at transplant, infection risk, and transplant-related variables, such as donor type, graft source, graft composition, are taken into account in calculating the starting dose. Moreover, the starting daily dose is approximately corrected for the patient’s age, according to the ward’s internal protocol (≥ 50 mg/kg in patients under six years of age, 30-50 mg/kg in older patients). We added a sentence in section 2.3, lines 111-115.
4) “They changed MMF dosage. How was the strategy of MMF dose change including timing of change and the efficacy of the dose change of MMF on GvHD?”
-We appreciate this referee’s question. Starting from the MMF initial daily dose, and tailoring MMF dosage based on the first AUC0-12, thereafter we performed AUC measurement of MPA weekly and we changed MMF dose until AUC measurement of MPA was within the defined range 30-60 µg/ml/h. We have elaborated in section 2.4, lines 129-132.
5) “The analytical methods’ performance including interday and intraday variability must be described”
-We appreciate this referee’s comment. We have included our reply in section 2.4, lines 120-126: “The limit of detection (LOD) was 0.4 µg/mL. The method revealed linearity between 0.4 µg/mL and 20 µg/mL (mean correlation coefficient, R2=0.998). Our laboratory found that the intra-day precision (RDS%) and accuracy (%) of the method at MPA plasma concentrations of 0.4, 4.0, 20.0 µg/mL was 6.91% and -0.50%, 6.49% and 0.40% and 5.57% and 0.75%, respectively; the corresponding inter-day precision (RDS%) and accuracy (%) was estimated to be 6.63% and -0.10%, 6.69% and -0.69% and 5.94% and 1.43%, respectively.”
6) “There are several typos in the manuscript”
-We appreciate this referee’s comment and we have improved our text.
7) “Please add the data of age in Table”
- We appreciate this referee’s comment. The age at transplant was already mentioned in Table 1 (please see the fifth line in the above-mentioned table).
Again, we want to thanks the reviewers for their insightful comments. Notice that all the corrections are clearly highlighted in yellow in the revised manuscript (we have mentioned related lines in this letter).
We hope the manuscript is now improved. However, we are fully available for other changes if requested.
Enclosed you can find a copy of the revised manuscript with highlights.
Sincerely,
Maximova Natalia
Reviewer 3 Report
Peer review on Carlone et al: "Area under the curve-based mycophenolate mofetil dosage..."
The manuscript by Carlone et al is reporting the incidences of acute and chronic GVHD following pediatric alloSCT, comparing a contemporary cohort treated with various pharmacokinetics (AUC)-guided mycophenolate mofetil (MMF)-based GVHD prophylaxis regimens with a historical cohort composed of SCT recipients treated with a variety of different other GVHD prophylactic regimen, including fixed dose MMF-based ones. It is shown that AUC-guided MMF dosing can increase the proportion of SCT recipients reaching a defined target AUC.
There are some major issues, partly even foiling the conclusion formulated in the study's title:
Both groups are very heterogeneous: both contain haplo-SCT and HLA-matched SCT. However, in the control group, haploSCT were obviously performed without PTCy but instead with MTX+MMF (and ATG, as mentioned in the text). This difference likely contributes to differences in GVHD frequencies. Another difference between study group and control group is calcineurin inhibitor backbone (TAC versus CSA). Furthermore, MMF-AUC based transpants are compared with a mixture of CSA/MTX and CSA/MTX/MMF based SCT.
Thus, the different GVHD incidences between study group and historical group cannot be attributed solely to the AUC measuring, but probably are the result of a mixture of different methods in both groups.
Therefore, groups should be better matched, leaving MMF fixed dose versus AUC-guided dosing as the sole principle difference. Multivariable testing would help by considering confounding effects, but therefore, cohorts need to be significantly larger.
As shown in table 2, initial MMF dose in the study group was 55.5 mg/kg, as compared to 30mg/kg in the historical group. The median final MMF dose in the study group was only slightly higher (59.2 mg/kg). Therefore, different GVHD outcomes may be the result of a higher MMF starting dose in the study group, as well as resulting from dose changes according to AUC findings.
It should be emphasized more clearly which novel insights the present study adds to the current state of evidence. For example, differences in MPA pharmacokinetics between various pediatric age groups have been described previously, as cited in the discussion. Which of the presented findings are novel?
Finally, a clinical conclusion should be formulated: do the findings suggest that PMA AUC should become routine? Or should higher MMF doses, compared to current standards, be evaluated for GVHD prophylaxis?
Specific issues:
Kaplan Meier (KM) curves for OS and PFS, as well as Cumulative Incidence curves for relapse and for non-relapse mortality should be shown. At least, these outcomes should be reported in the text.
KM-curves are inappropriate for the description of cumulative incidences of events that have competing risks (e.g., death without cGVHD as competing risk for cGVHD). An appropriate cumulative incidence function should be used instead.
The route of application of MMF is not indicated – was it i.v. in all patients?
CSA and TAC target trough levels are not indicated.
Significant previous contributions to the addressed field have been passed over, for example, Haentzschel et al: Targeting mycophenolate mofetil for graft-versus-host disease prophylaxis after allogeneic blood stem cell transplantation. Bone Marrow Transplant. 2008 Jul;42(2):113-20.
Minor issues:
Table 1, section Graft source, graft composition and GVHD prophylaxis: the lines are not clearly assigned, they are somewhat displaced
Author Response
Reply to Reviewer #3:
We want to thank the Reviewer for his thoughtful and insightful comments.
“Major issues”
1)“Both groups are very heterogeneous: both contain haplo-SCT and HLA-matched SCT. However, in the control group, haploSCT were obviously performed without PTCy but instead with MTX+MMF (and ATG, as mentioned in the text). This difference likely contributes to differences in GVHD frequencies”
- We agree with this comment. We checked the data inserted in Table 1 and corrected the number of transplants with Cy-post that were wrongly reported. 4 of 8 patients in the study group who underwent a haploidentical transplant were unable to undergo high-dose cyclophosphamide because of severe complications or pre-existing comorbidities. The small number of patients undergoing Cy-post GVHD prophylaxis did not lead to statistically significant variations compared with the historical group. We corrected Table 1.
2) “Another difference between study group and control group is calcineurin inhibitor backbone (TAC versus CSA). Furthermore, MMF-AUC based transplants are compared with a mixture of CSA/MTX and CSA/MTX/MMF based SCT” and “Thus, the different GVHD incidences between study group and historical group cannot be attributed solely to the AUC measuring, but probably are the result of a mixture of different methods in both groups”
- We appreciate this referee’s comment. We have already reported in the original manuscript, section Discussion, that different prophylactic regimens used for GvHD prophylaxis in the historical group could be responsible for the higher detected incidence of acute and chronic GvHD in this cohort of patients. However, this referee’s comment allowed us to think better about heterogeneity of groups and to highlight the fact that the heterogeneity allowed us to obtain a larger sample size for statistical analysis, despite limitations of this choice. We have improved our reply in section 4, lines 54-57.
Furthermore, we agree that the calcineurin inhibitor change (CsA versus TAC) certainly affected GvHD incidence. But we are always faced with the fact that the numerical limit of the sample precludes a multivariate analysis. In the future we hope to carry out a multicenter project with centers that use cyclosporine and MMF in prophylaxis to rule out the problem of heterogenicity and sample size.
3) “Therefore, groups should be better matched, leaving MMF fixed dose versus AUC-guided dosing as the sole principle difference. Multivariable testing would help by considering confounding effects, but therefore, cohorts need to be significantly larger”
-We agree with the referee’s comment and we acknowledged this as a limit. Our study is retrospective, and we cannot change our sample size, and any further fragmentation of our population would make any statistical comparison impossible. We added a sentence with an appropriate comment in section 4, lines 54-57.
4) “As shown in table 2, initial MMF dose in the study group was 55.5 mg/kg, as compared to 30mg/kg in the historical group. The median final MMF dose in the study group was only slightly higher (59.2 mg/kg). Therefore, different GVHD outcomes may be the result of a higher MMF starting dose in the study group, as well as resulting from dose changes according to AUC findings”
- We appreciate this referee’s comment. Our study showed the relevance of MPA AUC0-12 monitoring in order to obtain, early on, the tailored dose of MMF, which is clearly influenced by differences in MPA pharmacokinetics between various pediatric age groups. Practically speaking, this allowed us to obtain better outcomes concerning GvHD incidence. We have included our reply in section 4, lines 32-36.
5) “It should be emphasized more clearly which novel insights the present study adds to the current state of evidence. For example, differences in MPA pharmacokinetics between various paediatric age groups have been described previously, as cited in the discussion. Which of the presented findings are novel?”
- We appreciate this referee’s comment. We added a sentence in section 4, line 36-40: “Furthermore, our results add data to available literature by showing relevant difference in MPA pharmacokinetics between patients under or over six years of age. This new cut-off age should manage the choice of different starting MMF daily doses, compared to current standards, for GvHD prophylaxis in pediatric patients, in addition to AUC measurement of MPA.”
6) “Finally, a clinical conclusion should be formulated: do the findings suggest that PMA AUC should become routine? Or should higher MMF doses, compared to current standards, be evaluated for GVHD prophylaxis?”
- We appreciate this referee’s comment. We added the following sentence in section 5, lines 61-64: “therefore, AUC measurement of MPA should be taken into consideration in these patients, as well as tailored initial MMF daily doses with respect to different pediatric age groups, in order to obtain better outcomes through early achievement of the most effective prophylactic dose.”
“Specific issues”
1) “Kaplan Meier (KM) curves for OS and PFS, as well as Cumulative Incidence curves for relapse and for non-relapse mortality should be shown. At least, these outcomes should be reported in the text”
- We fully agree with the Reviewer’s comment and we have included our reply in section 3.3, lines 174-179. Moreover, we have added the new Figure 2 showing KM curves for overall survival, as well as Figure 3 showing the Cumulative Incidence curve for relapse mortality.
In addition, since we have not found statistically significant differences in the overall survival and progression-free survival between the study and the historical group, we added a comment in section 4, lines 41-46.
2) “KM-curves are inappropriate for the description of cumulative incidences of events that have competing risks (e.g., death without cGVHD as competing risk for cGVHD). An appropriate cumulative incidence function should be used instead”
- We fully agree with the Reviewer’s comment and we deleted the Figure 1 showing KM curves for overall incidence of acute and chronic GvHD.
3) “The route of application of MMF is not indicated – was it i.v. in all patients?”
4) “CSA and TAC target trough levels are not indicate”.
- We appreciate this referee’s comment and we have included our reply in section 2.3, lines 105-110: “Specifically, CsA was administered intravenously at a dose of 1 to 3 mg/kg/d for the first 21 days and subsequently orally at a dose of 6 mg/kg/d. Initial target trough level was set at 250-350 ng/mL and, later, during oral administration at 200-250 ng/mL. On the other hand, FK506 was administered intravenously at a dose of 0.03 mg/kg/d with initial target concentration setting at 15-20 ng/mL, while, during oral administration, the target trough level was set at 10-15 ng/mL”.
5) “Significant previous contributions to the addressed field have been passed over, for example, Haentzschel et al: Targeting mycophenolate mofetil for graft-versus-host disease prophylaxis after allogeneic blood stem cell transplantation. Bone Marrow Transplant. 2008 Jul;42(2):113-20”
-We appreciate this referee’s comment. We know and we esteem this insightful study; furthermore, it has partly inspired our study involving pediatric patients undergoing allo-HSCT. However, we have not mentioned it before because our MPA AUC range was slightly different from the one proposed in the above-mentioned study. Still, we have improved our references including it and we have reported it section 2.4, line 130 (reference 11).
Minor issues
“Table 1, section Graft source, graft composition and GVHD prophylaxis: the lines are not clearly assigned, they are somewhat displaced”
-We fully agree with the Reviewer’s comment and we have improved the above-mentioned mistakes; see section 3.1, Table 1.
Again, we want to thanks the reviewers for their insightful comments. Notice that all the corrections are clearly highlighted in yellow in the revised manuscript (we have mentioned related lines in this letter).
We hope the manuscript is now improved. However, we are fully available for other changes if requested.
Enclosed you can find a copy of the revised manuscript with highlights.
Sincerely,
Maximova Natalia
Round 2
Reviewer 2 Report
They responded to my concerns.
Author Response
Thank you.
Reviewer 3 Report
Peer review on revised manuscript 02-01-2021
The manuscript has been substantially improved by adding the suggested additional information, particularly on details regarding immunosuppressive drug dosing. Also, study limitations are now discussed more appropriately.
A) However, my major issue with the study is still remaining: It is not possible to conclude superiority of AUC directed MMF dosing from a comparison of AUC-guided MMF-based transplants with a historical cohort of transplants that contains a significant number of MTX-based ones (without any use of MMF). Therefore, although particularly the control group will become significantly smaller, all primary and secondary outcomes should be recalculated for more homogeneous cohorts:
1) The four PTCy based transplants should be excluded from the study cohort, because addition of PTCy possibly has a greater impact on GVHD than does AUC-directed MMF dosing (leaving 51 instead of 55 patients in the study cohort).
2) MTX-based transplants need to be excluded from the control cohort, because they are entirely irrelevant for a comparison of standard MMF vs AUC-guided MMF dosing. Therefore, 25 transplants will remain in the control group.
GVHD prophylaxis regimens termed “other” should be specified, and should remain in the analysis only if they are MMF-based.
B) It is questionable whether DLI-induced GVHD should be included in the analysis, because they are causally closer related to the intervention of DLI than to the primary immunosuppressive regimen. At least, it should be stated how many DLI-induced GVHD events were included, and whether any immunosuppression (particularly with MMF) was still ongoing at the time of DLI.
With the above modifications, the study will be better focused on MMF, and conclusions can be made with higher validity. However, even then, the hypothesis-generating character of the study needs to be considered, and the title (e.g., “may contribute to the reduction…” instead of “reduced”, as well as the conclusions in the text, need to be formulated in a less definitive diction.
Author Response
January 7th, 2021
Revision: R2 of the manuscript jcm-1025183 entitled “Area under the curve-based mycophenolate mofetil dosage reduced the incidence of graft-versus-host disease after allogeneic hematopoietic cell transplantation in pediatric patients”.
The manuscript was revised and below we have addressed point to point the issues raised in the comments.
Reply to Reviewer #3:
We want to thank the Reviewer for his thoughtful and insightful comments.
- A) “However, my major issue with the study is still remaining: It is not possible to conclude superiority of AUC directed MMF dosing from a comparison of AUC-guided MMF-based transplants with a historical cohort of transplants that contains a significant number of MTX-based ones (without any use of MMF). Therefore, although particularly the control group will become significantly smaller, all primary and secondary outcomes should be recalculated for more homogeneous cohorts:”
1) The four PTCy based transplants should be excluded from the study cohort, because addition of PTCy possibly has a greater impact on GVHD than does AUC-directed MMF dosing (leaving 51 instead of 55 patients in the study cohort).
2) MTX-based transplants need to be excluded from the control cohort, because they are entirely irrelevant for a comparison of standard MMF vs AUC-guided MMF dosing. Therefore, 25 transplants will remain in the control group
- We appreciate and we fully understand this referee’s comment. We have updated data included in the Table 1 (please, see Table 1, section 3.1); furthermore, we have recalculated Kaplan-Meier curve for overall survival and Cumulative Incidence curve for relapse-related mortality including 51 patients in the study group and 25 patients in the control group (please, see Figure 2 and Figure 3, respectively-section 3.3).
…..
“GVHD prophylaxis regimens termed “other” should be specified, and should remain in the analysis only if they are MMF-based”.
- We appreciate this referee’s comment. We have added these data in the Table 1 (please, see Table 1, section 3.1)
- B) It is questionable whether DLI-induced GVHD should be included in the analysis, because they are causally closer related to the intervention of DLI than to the primary immunosuppressive regimen. At least, it should be stated how many DLI-induced GVHD events were included, and whether any immunosuppression (particularly with MMF) was still ongoing at the time of DLI.
- We fully agree with this referee’s comment. We excluded patients with DLI-induced GVHD from our analysis, as none of them took immunosuppressants during DLI treatment. We redid the Figure 1 (please, see Figure 1, section 3.2). We didn't have to further reduce either the study group or the control group, because all the above-mentioned patients were already eliminated from our cohort by answering point A 1-2.
Furthermore, we have changed the title, as suggested (please, see the title of the manuscript) and we have reformulated our conclusions in a less definitive manner (please, see section 5, line 60).
Again we want to thanks the above-mentioned reviewer for their insightful and efficient comments. Notice that all the corrections are clearly highlighted in yellow in the revised manuscript (we have mentioned related sections and lines in this letter).
We hope the manuscript is now improved. However, we are fully available for other changes, if requested.
Enclosed you can find a copy of the revised manuscript with highlights.
Sincerely,
Maximova Natalia
Bone Marrow Transplant Unit,
Institute for Maternal and Child Health – IRCCS “Burlo Garofolo”, 34137 Trieste, Italy
Tel: +39 040 3785276/565; Fax: +39 040 3785494
natalia.maximova@burlo.trieste.it
